# The contrasting phylodynamics of human influenza B viruses

Dhanasekaran Vijaykrishna[1,2,3]*, Edward C Holmes[4], Udayan Joseph[1],
Mathieu Fourment[4], Yvonne CF Su[1], Rebecca Halpin[5], Raphael TC Lee[6], Yi-Mo Deng[3],
Vithiagaran Gunalan[6], Xudong Lin[5], Timothy B Stockwell[5], Nadia B Fedorova[5],
Bin Zhou[5], Natalie Spirason[3], Denise Kühnert[7], Veronika Bošková[8], Tanja Stadler[8],
Anna-Maria Costa[9], Dominic E Dwyer[10], Q Sue Huang[11], Lance C Jennings[12],
William Rawlinson[13], Sheena G Sullivan[3,14], Aeron C Hurt[3,14],
Sebastian Maurer-Stroh[6,15,16], David E Wentworth[5], Gavin JD Smith[1,3,17]*,
Ian G Barr[3,18]

[1]Duke-NUS Graduate Medical School, Singapore, Singapore; [2]Yong Loo Lin
School of Medicine, National University of Singapore, Singapore, Singapore;
[3]World Health Organisation Collaborating Centre for Reference and Research on
Influenza, Peter Doherty Institute for Infection and Immunity, Melbourne,
Australia; [4]Marie Bashir Institute for Infectious Diseases and Biosecurity, University
of Sydney, Sydney, Australia; [5]J Craig Venter Institute, Rockville, United States;
[6]Bioinformatics Institute, Agency for Science, Technology and Research,
Singapore, Singapore; [7]Department of Environmental Systems Science,
Eidgenössische Technische Hochschule Zürich, Zürich, Switzerland; [8]Department
of Biosystems Science and Engineering, Eidgenössische Technische Hochschule
Zürich, Zurich, Switzerland; [9]Royal Children's Hospital, Parkville, Australia; [10]Centre
for Infectious Diseases and Microbiology Laboratory Services, Westmead Hospital and
University of Sydney, Westmead, Australia; [11]Institute of Environmental Science and
Research, National Centre for Biosecurity and Infectious Disease, Upper Hutt, New
Zealand; [12]Microbiology Department, Canterbury Health Laboratories, Christchurch,
New Zealand; [13]Virology Division, SEALS Microbiology, Prince of Wales Hospital, Sydney,
Australia; [14]School of Population and Global Health, University of Melbourne, Melbourne,
Australia; [15]School of Biological Sciences, Nanyang Technological University, Singapore,
Singapore; [16]National Public Health Laboratory, Communicable Diseases Division,
Ministry of Health, Singapore, Singapore; [17]Duke Global Health Institute, Duke
University, Durham, United States; [18]School of Applied Sciences and Engineering,
Monash University, Churchill, Australia

*For correspondence: vijay.
dhanasekaran@duke-nus.edu.sg
(DV); gavin.smith@duke-nus.edu.
sg (GJDS)

Reviewing editor: Richard A
Neher, Max Planck Institute for
Developmental Biology,
Germany

**Abstract** A complex interplay of viral, host, and ecological factors shapes the spatio-temporal
incidence and evolution of human influenza viruses. Although considerable attention has been paid
to influenza A viruses, a lack of equivalent data means that an integrated evolutionary and
epidemiological framework has until now not been available for influenza B viruses, despite their
significant disease burden. Through the analysis of over 900 full genomes from an epidemiological
collection of more than 26,000 strains from Australia and New Zealand, we reveal fundamental
differences in the phylodynamics of the two co-circulating lineages of influenza B virus (Victoria and
Yamagata), showing that their individual dynamics are determined by a complex relationship
between virus transmission, age of infection, and receptor binding preference. In sum, this work
identifies new factors that are important determinants of influenza B evolution and epidemiology.

**eLife digest** To develop new therapies against infections caused by a virus, it is important to understand the virus's history—where, when, and why it has caused disease and how it has changed over time. For example, new human strains of the influenza type A virus originate from strains that infect animals and rapidly can become common in human populations. In contrast, influenza type B virus strains almost exclusively infect humans and are continuously present in human populations. Both types have a detrimental impact on global health, but the type B viruses are less well understood, partly because outbreaks have not been as extensively documented.

Vijaykrishna et al. have now investigated the history of the two strains of the influenza type B virus—called Victoria and Yamagata—that currently circulate in humans. To do this, they inspected the genetic sequences of 908 viruses taken from samples of confirmed type B infections collected across Australia and New Zealand over 13 years.

Individual virus particles of the same strain have genetic sequences that are very similar, but not completely identical. Vijaykrishna et al. showed that the diversity of the genetic sequences from the Victoria strain fluctuated between seasons, and particular genetic variants of Victoria only persisted in the population for 1–3 years. This indicates that Victoria viruses are under a lot of pressure to evolve, which results in so-called 'bottlenecks' whereby only the viruses carrying particular varieties of genetic sequence survive. This fluctuating pattern resembles that of the better-understood type A seasonal flu strain H3N2.

On the other hand, there was little change in the genetic diversity of the Yamagata strains sampled over the same 13-year period. The Yamagata viruses have diversified to a greater extent and several different 'varieties' of the virus tend to circulate together for long periods of time. For example, the three varieties of Yamagata virus circulating in 2013 evolved from a common parent virus that was circulating around 10 years ago.

Vijaykrishna et al. found that between disease outbreaks, there was greater variation in the ability of Victoria viruses to be transmitted in humans, but that they were generally more easily transmitted than the Yamagata viruses. Victoria viruses tend to infect younger patients than Yamagata viruses, which is thought to be due to differences in the molecules that help the viruses enter the cells of the respiratory tract.

These findings suggests that it might be possible to eradicate the more slowly evolving influenza B Yamagata virus through mass vaccination programs using existing vaccines. This would then allow researchers to focus on developing effective vaccines to target the other strains of influenza virus.

## Introduction

In addition to two subtypes of influenza A virus (H1N1 and H3N2), two lineages of influenza B viruses co-circulate in humans and cause seasonal influenza epidemics (*Klimov et al., 2012*). Influenza B causes a significant proportion of influenza-associated morbidity and mortality, and in some years is responsible for the major disease burden (*Burnham et al., 2013*; *Paul Glezen et al., 2013*). Although type A and B influenza viruses are closely related and have similarities in genome organization and protein structure (*McCauley et al., 2012*), they exhibit important differences in their ecology and evolution (*Chen and Holmes, 2008*; *Tan et al., 2013*). While new influenza A viruses periodically emerge from animal reservoirs to become endemic in humans (*Neumann et al., 2009*; *Smith et al., 2009*), influenza B viruses, first recognized in 1940, have circulated continuously in humans alongside influenza A viruses and are presumably derived from a single, as yet unknown, source (*Francis, 1940*; *Chen and Holmes, 2008*). Unlike influenza A viruses that can infect a wide range of species, influenza B infections are almost exclusively restricted to humans with only sporadic infections reported in wildlife (*Osterhaus et al., 2000*; *Bodewes et al., 2013*). While the evolutionary and epidemiological dynamics of human influenza A H1N1 and H3N2 viruses have been well documented at the global scale (*Rambaut et al., 2008*; *Russell et al., 2008*; *Bedford et al., 2010*; *Bahl et al., 2011*), the equivalent dynamics of the two influenza B virus lineages—B/Yamagata/16/88-like and B/Victoria/2/87-like, henceforth termed the Yamagata and Victoria viruses—are poorly understood.

Human influenza A H3N2 viruses exhibit limited genetic diversity at individual time-points due to periodic bottlenecks caused by strong selection—known as 'antigenic drift'—in the hemagglutinin (HA) and neuraminidase (NA) genes. This results in an HA phylogenetic tree with a characteristic slender 'trunk' (*Fitch et al., 1997*) appearance (*Figure 1A*). H3N2 viruses also exhibit strong seasonal fluctuations in genetic diversity in temperate climate regions (such as Australia and New Zealand) (*Rambaut et al., 2008*), mainly due to the local extinction of viral lineages at the end of each influenza season (*Rambaut et al., 2008*). A similar but weaker evolutionary pattern is observed in the seasonal H1N1 viruses that have circulated in humans for the majority of the previous century (1918–1957 and 1977–2009), with short-term co-circulation of diverging virus populations (*Nelson et al., 2008b*) (*Figure 1B*). The pandemic H1N1 (H1N1pdm09) viruses have to date also only exhibited limited antigenic evolution since they emerged in 2009 (*Figure 1C*). In contrast, influenza B viruses are currently composed of two distinct lineages (Victoria and Yamagata) (*Kanegae et al., 1990*; *Rota et al., 1990*) (*Figure 1D*) that diverged approximately 40 years ago and which have since co-circulated on a global scale, despite frequent reassortment among them (*Chen and Holmes, 2008*). Although the HA genes of influenza B viruses are thought to exhibit lower rates of evolutionary change (nucleotide substitution) than both influenza A viruses (*Ferguson et al., 2003*; *Chen and Holmes, 2008*; *Bedford et al., 2014*), their antigenic characteristics are not well understood.

The advent of global influenza surveillance and full genome sequencing over the past decade has shown that seasonal epidemic outbreaks of each influenza type are caused by the stochastic introduction of multiple virus lineages (*Nelson et al., 2008a*) and that the patterns of seasonal oscillation vary between temperate and tropical regions (*Rambaut et al., 2008*). Population genetic analysis (*Rambaut et al., 2008*), consistent with epidemiological data (*Goldstein et al., 2011*), suggests that the H3N2 and H1N1 subtypes of influenza A virus compete with each other resulting in the epidemic dominance of a single subtype. However, it is unclear whether the same dynamic patterns can be extended to influenza B viruses, or why the Victoria and Yamagata lineages have co-circulated for such an extended time period.

To understand the evolutionary and epidemiological dynamics of influenza B virus, we generated the full genomes of 908 influenza B viruses selected from over 26,000 laboratory confirmed influenza B cases in children and adults aged from birth to 102 years sampled during 2002–2013 in eastern

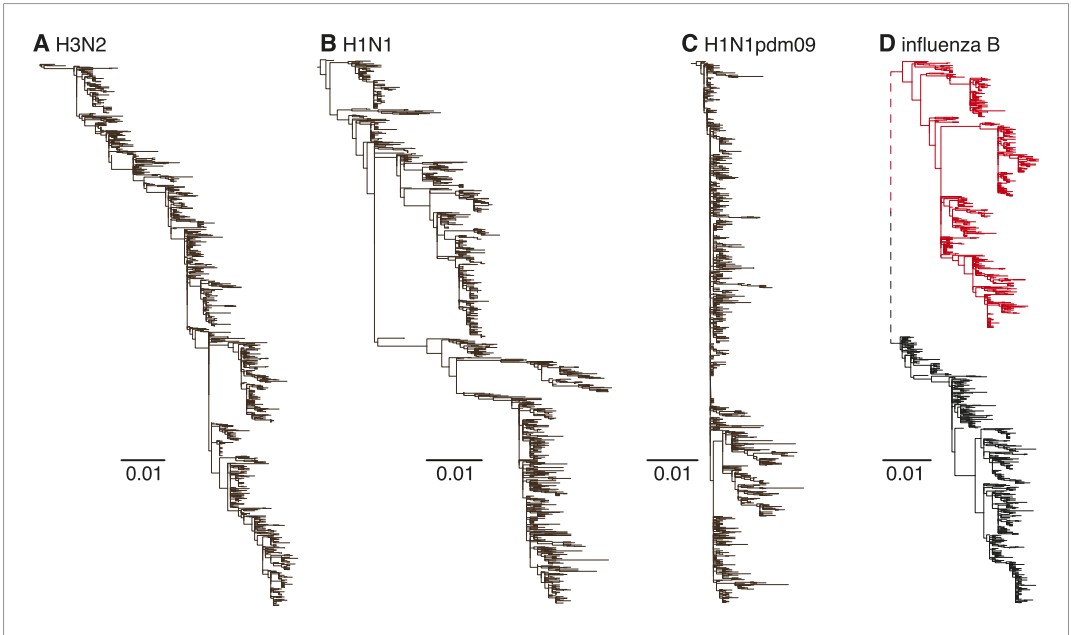

**Figure 1**. Evolutionary dynamics of human influenza A and influenza B Victoria and Yamagata viruses. Evolution of the HA genes of influenza A H3N2 virus, 2002–2013, (**A**), H1N1 virus, 1998–2009 (**B**), H1N1pdm09 virus, 2009–2013 (**C**), and influenza B Yamagata (red) and Victoria (black) lineage viruses, 2002–2013 (**D**). All phylogenetic trees were generated using approximately 1200 randomly selected full-length gene sequences sampled during 12 years.

Australia (Queensland, n = 275; New South Wales, n = 210; and Victoria, n = 207) and New Zealand (n = 216) (*Figure 2*). These regions were selected because influenza surveillance was well established and continuous during the sampling period and included the co-circulation and periodic dominance of influenza A and both influenza B virus lineages. Of note is that the influenza B virus strain used for vaccination in the region did not match the dominant circulating strain during 7 of the 13 years studied (*Figure 2B*). Our overall aim was to integrate, for the first time, data obtained from genetic, epidemiological, and immunological sources to better understand the evolution and interaction of these two lineages of influenza B virus.

## Results and discussion

### Population dynamics of influenza B virus

We used the HA segment of both lineages to contrast their phylodynamics. First, to assess the changing patterns of genetic diversity of the two influenza B virus lineages in relation to their evolutionary histories, we used a flexible coalescent-based demographic model (*Minin et al., 2008*), which revealed stark differences in the epidemiological dynamics of the Victoria and Yamagata lineages (*Figure 3A,B*). Whereas the Victoria lineage experienced strong seasonal fluctuations in

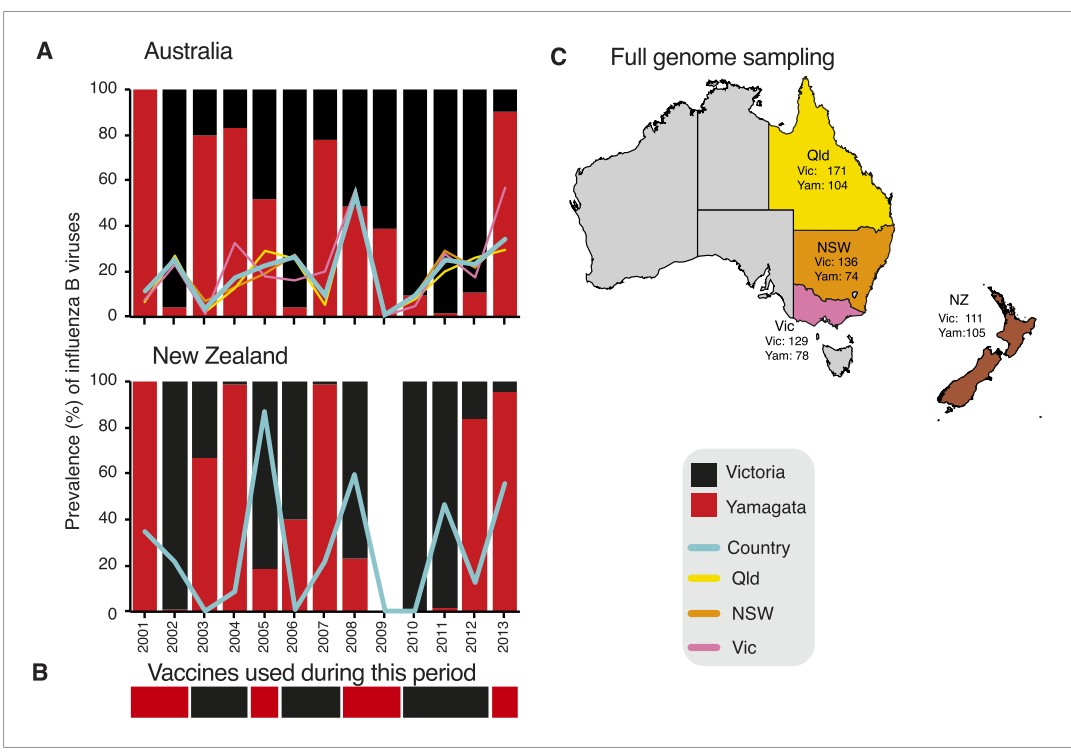

**Figure 2**. Influenza B virus lineages in Australia and New Zealand, 2001–2013 and source of full genomes. Percentage prevalence of influenza B viruses isolated from the three eastern Australian states and New Zealand (**A**). Coloured lines represent the proportion of influenza viruses typed as influenza B in each country (blue) and each of the eastern Australian states; Queensland (yellow), New South Wales (orange), and Victoria (pink). Bars represent the percentage prevalence of Victoria (black) and Yamagata (red). Data based on National Notifiable Diseases Surveillance system (NNDSS) for Australia and Environmental Science and Research (ESR) for New Zealand. The lineage of representative influenza B virus strains used in the trivalent influenza vaccine during these years in both countries (**B**). Excluding the years 2003 and 2009, influenza B viruses represented on average 24.6% (range 9.5–53.7%) and 31.5% (range 0.5–86.9%) of laboratory confirmed influenza viruses from Australia and New Zealand, respectively. The percentage of circulating influenza viruses that were influenza B was significantly lower in 2003 (AUS, 3.4%) and 2009 (AUS, 0.8%) than in other years, due to the dominance of a new H3N2 variant (A/Fujian/412/2002-like) in 2003 and the emergence of the H1N1 pandemic in 2009. Source of full genomes of Victoria and Yamagata viruses (**C**).

relative genetic diversity, little change was observed over the same time period for the Yamagata lineage, and these observations were not heavily affected by differences in sampling density (*Figure 3—figure supplement 1*). While the almost invariant relative genetic diversity of the Yamagata lineage resembled that of seasonal H1N1 viruses (*Figure 3D*), the stark and almost annual changes of diversity in the Victoria lineage were similar to those observed for H3N2 virus (*Figure 3C*); although H3N2 viruses exhibited a greater frequency of oscillations than those estimated for Victoria lineage viruses. The strong seasonal fluctuations in diversity observed for Victoria lineage suggest that this lineage experiences strong bottlenecks between seasons similar to those described for H3N2 viruses (*Bedford et al., 2011*; *Zinder et al., 2013*), whereas the almost invariant relative genetic diversity for Yamagata suggests the continuous co-circulation of multiple lineages.

Marked differences between the Victoria and Yamagata lineages were apparent in phylogenetic trees of the HA (*Figure 4*). The phylogenetic analysis of the HA genes showed that the Victoria lineage was characterized by a single prominent tree 'trunk', with side branches that circulated for short periods of time (1–3 years) (*Figure 4*). This evolutionary pattern parallels that observed for seasonal H3N2 viruses and is indicative of frequent selective bottlenecks due to the serial replacement of circulating strains, as would be expected under continual antigenic drift (*Grenfell et al., 2004*). In contrast, greater diversification was observed for the Yamagata lineage, with multiple clades co-circulating for extensive periods of time (*Figure 4*). For example, the three clades of Yamagata viruses circulating in 2013 diverged approximately 10 years ago, again paralleling the evolutionary pattern seen in seasonal H1N1 viruses. These patterns are clearly identifiable in the genealogical diversity skyline (*Figure 4*) in which the average time to common ancestor between contemporaneous samples fluctuated from 0 to <5 years for Victoria lineage, except during 2010 and 2011 where the

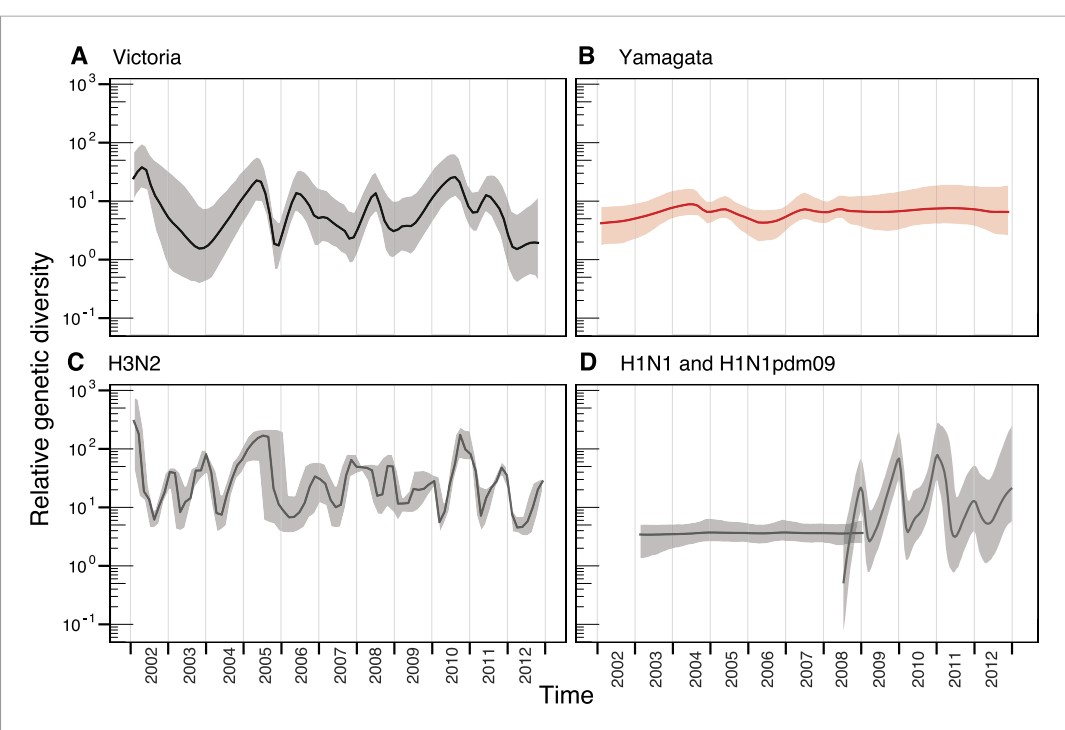

**Figure 3**. Population dynamics of genetic diversity in Australia and New Zealand. The relative genetic diversity of the HA segments of influenza B Victoria (**A**), Yamagata (**B**) and influenza A H3N2 (**C**), and H1N1 2003–2008 and H1N1pdm09 2009–2013 viruses (**D**), isolated in Australia and New Zealand using the Gaussian Markov Random Field (GMRF) model.
The following figure supplement is available for figure 3:

**Figure supplement 1**. Effect of sampling on the population dynamics of Influenza B virus.

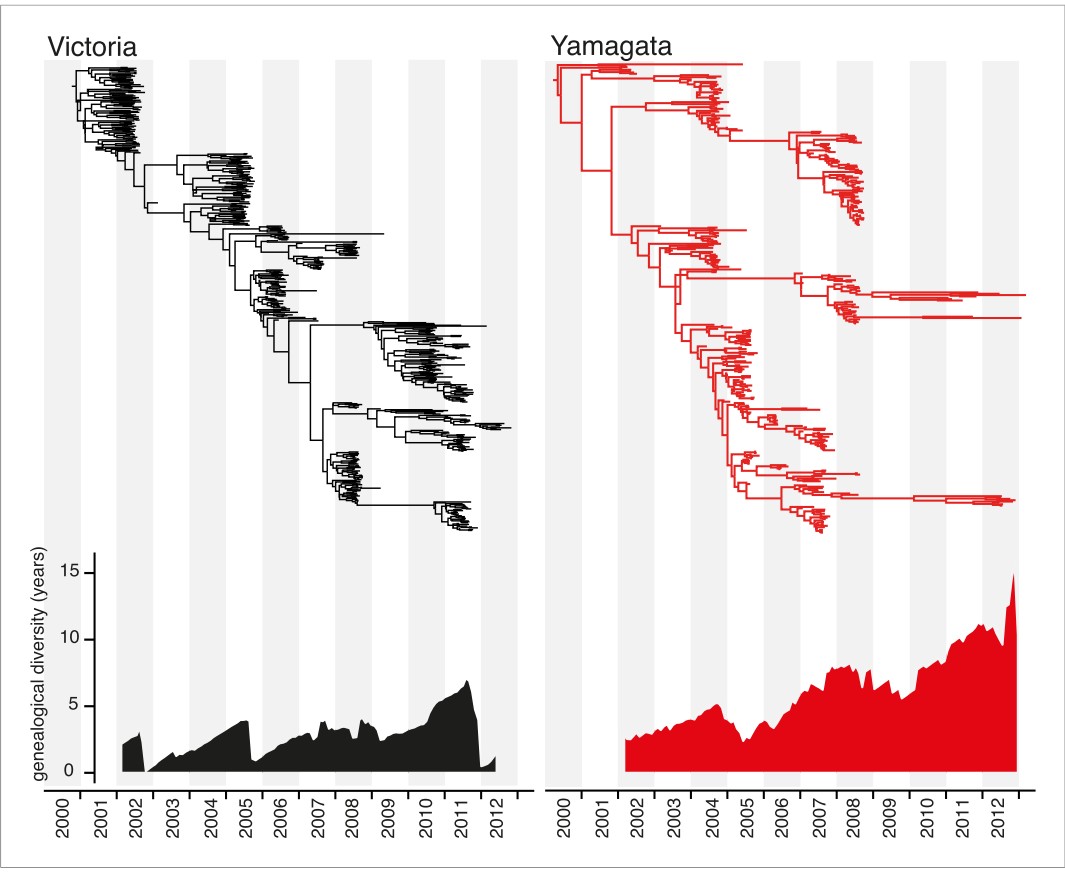

**Figure 4**. Evolution of the hemagglutinin genes of influenza B viruses. Phylogenetic relationship of the HA genes of influenza B Victoria (black) and Yamagata (red) lineage viruses inferred using the uncorrelated lognormal relaxed clock model. Genetic diversity through time was estimated by averaging the pairwise distance in time between random contemporaneous samples with a 1-month window on the same dated Maximum clade credibility (MCC) trees.

genealogical diversity marginally increased to 7 years. In contrast, the genealogical diversity of Yamagata was consistently greater and gradually increased during the sampling period. The maintenance of genetic diversity through epidemic peaks and troughs as described for Yamagata (*Figure 3B*) is expected to result in the gradual increase of divergence times of contemporaneous samples.

### Transmission dynamics of influenza B virus

As each seasonal influenza epidemic provides important information on the epidemiological characteristics of both influenza B virus lineages, we utilized a birth–death susceptible-infected-removed (BDSIR) (*Kühnert et al., 2014*) phylodynamic model that simultaneously co-estimates seasonal phylogenies along with the basic reproductive number, $R_0$, for each lineage. However, because the infected population contains both susceptible and non-susceptible hosts we report the effective reproductive number, $R_e$. This analysis showed a greater variation in $R_e$ (median values, 1.1–1.3) between epidemics caused by the Victoria lineage, whereas the $R_e$ of Yamagata epidemics, were generally lower, varied only slightly, around 1.1 (1.08–1.14) (*Figure 5A*), indicating greater heterogeneity in transmission between seasons for Victoria viruses. Years in which both influenza viruses co-circulated in sufficient numbers (2005 and 2008) offer a chance for direct comparison of their phylodynamics. Both lineages transmitted with nearly equal force in 2005, whereas in 2008 the median $R_e$ of 1.27 (95% highest posterior density [HPD] of 1.18–1.37) estimated for the Victoria lineage was significantly greater than that of Yamagata at 1.11 (95% HPD 1.05–1.17). Analysis of the

cumulative number of all influenza B positive cases through time for each season (*Figure 5B*) reveals significant differences in the exponential growth phase between the lineages, where Victoria lineage exhibited significantly higher initial growth rate resulting in a faster epidemic with larger number of infections. These results also show that in 2008 the Victoria lineage exhibited a significantly faster growth rate, in correlation with the high $R_e$, coinciding with the year in which a new antigenic variant of the Victoria lineage was first detected (B/Brisbane/60/2008-like viruses) in Australia and New Zealand. This antigenic variant emerged as the globally dominant influenza B strain in the following years and has been continuously recommended (2009–2015) as the influenza B vaccine component since that period in both the Northern and Southern Hemispheres (*Klimov et al., 2012*).

The BDSIR model assumes a closed epidemic, but the large-scale phylogenies generated using all available global data indicated that each of the annual epidemics were caused by the introduction of multiple viral lineages that went extinct locally by the end of the seasonal epidemic (data not shown). We therefore investigated the effect of virus migration on the estimates of $R_e$. First, we identified lineages that conformed to the assumption of a closed epidemic (i.e., lineages resulting from a single introduction into Australia and New Zealand) and with a sufficiently large local transmission for analysis (i.e., Victoria lineage viruses in 2005, 2006 and 2008). An independent estimation of $R_e$ for each of these lineages produced a minor but non-significant variation to those observed for the entire epidemic (*Figure 5—figure supplement 1B*), indicating that, on average, the $R_e$ estimates for lineages resulting from multiple introductions were similar. Next, we used a continuous-time Markov chain (CTMC) phylogeographic process (*Minin and Suchard, 2008*) to estimate the number of migration events into and from Australia and New Zealand during the same period (*Figure 6*). This indicated that the number of introductions per year was greater for the Yamagata lineage (15–22, mean state transition count in all years) than for Victoria (3–8, except 16 and 14 during 2010 and 2011, respectively) (*Figure 6*), further suggesting an inverse relationship between $R_e$ (*Figure 5*) and the number of introduction events. Indeed, our results show that introductions of viruses with greater transmission efficiency (i.e., high $R_e$), such as Victoria/2008, resulted in the epidemic dominance of such single strains, whereas epidemics of the Yamagata lineage with lower $R_e$ values likely resulted in slower and shorter transmission chains with reduced competition, in turn allowing the co-circulation (and detection) of multiple introduced lineages. Additionally, we identified that, combined, Australia and New Zealand were net importers of influenza viruses, except during 2002 and 2008 when the net export of the Victoria lineage was similar to the import observed during the

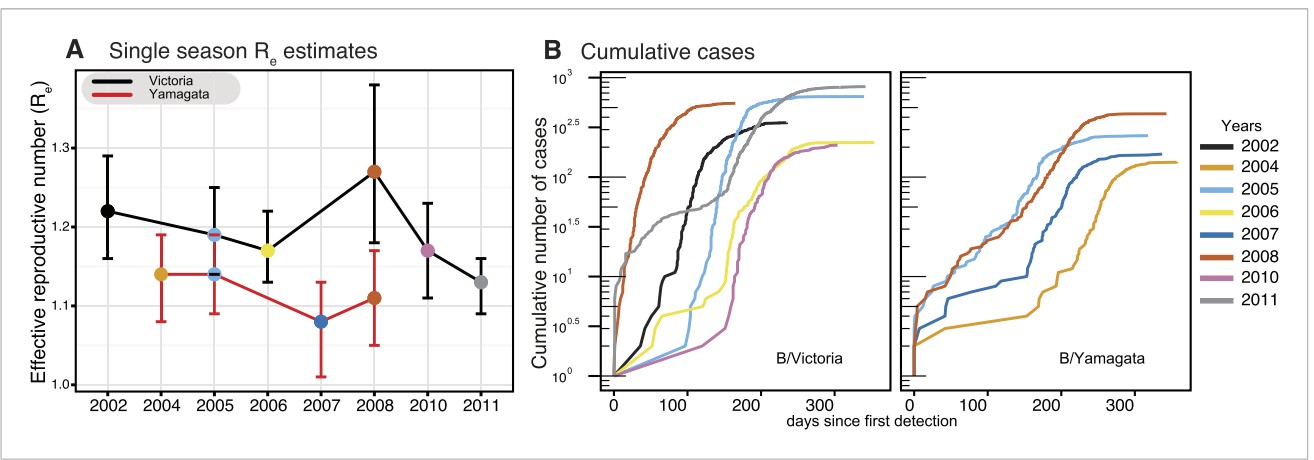

**Figure 5**. Phylodynamics and cumulative cases of influenza B viruses. Effective reproductive number ($R_e$) of influenza B Victoria (black) and Yamagata (red) viruses (of the HA data set) estimated for single epidemics (median and 95% highest posterior density (HPD) values) during years with sufficient number of sequences estimated using the BDSIR model (**A**). The cumulative number of cases from all influenza B virus positive samples for each of these years (**B**).

The following figure supplement is available for figure 5:

**Figure supplement 1**. Estimates of $R_e$ with various $S_0$ values.

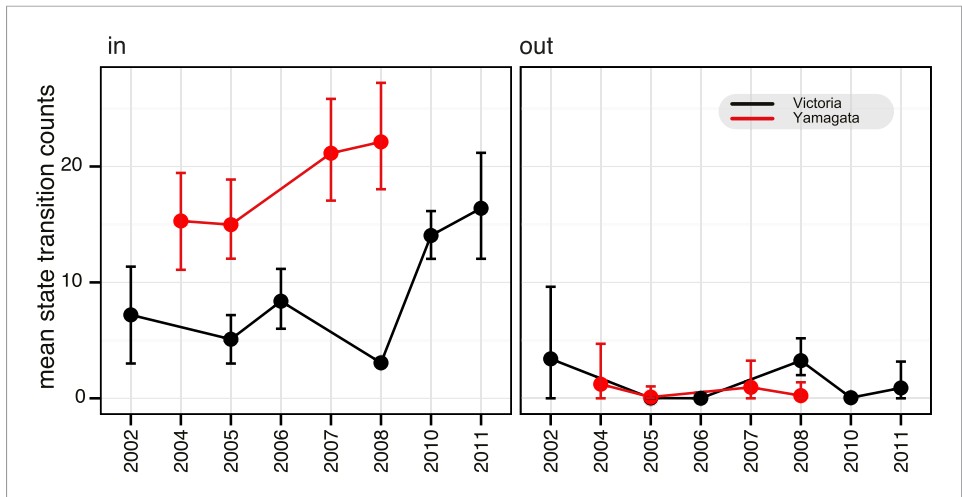

**Figure 6**. Estimation of migration of influenza B viruses into and out of Australia and New Zealand. Estimated counts of import and export of Victoria (black) and Yamagata (red) between Australia and New Zealand and rest of the world, using the HA gene data set. Error bars represent the 95% highest posterior density (HPD) values of each point.

same years (*Figure 6*). The higher transmission rate for Victoria/2008 viruses (i.e., B/Brisbane/60/2008-like viruses) may have also caused the successful seeding of these viruses globally (as described above). Taken together, our results support the concept of a global metapopulation seeding subsequent epidemics elsewhere (*Bedford et al., 2010*; *Bahl et al., 2011*), provided the virus is transmitted efficiently as observed during 2008 in this study.

## Genome-wide evolutionary dynamics of influenza B viruses

To understand the genome-wide evolutionary dynamics of the two influenza B virus lineages, we inferred temporal changes in genetic diversity for all remaining gene segments (*Figure 7*). These analyses showed that the patterns observed for the NA and internal gene segments were similar to those observed for the HA genes described above. The single exception was the NP genes of both lineages where substantial differences occurred throughout their history. During 2002–2007, the peaks of relative genetic diversity of the Victoria NP was higher than all remaining gene segments following which this lineage was not identified in our surveillance, whereas the Yamagata NP showed additional peaks during 2010 and 2011 that corresponded to the NP peaks observed for the Victoria genes.

As genomic reassortment impacts levels of genetic diversity, we conducted phylogenetic analyses of all eight genome segments of the 908 viruses. Comparison of these phylogenies revealed frequent reassortment within the two lineages of influenza B virus (data not shown) and a few instances of reassortment between them (*Figure 8*). During the sampling period, the Victoria lineage HA gene repeatedly acquired internal gene segments from Yamagata lineage viruses to form novel reassortants. In particular, during 2004 a subpopulation (approximately 15%) of Victoria-like viruses acquired all internal gene segments (PB2, PB1, PA, NP, MP, and NS) from the Yamagata lineage viruses. Interestingly, all remaining inter-lineage reassortment events of the Victoria HA lineages involved the acquisition of the Yamagata NP gene during 2007 and 2008 (*Figure 8E*), which resulted in the extinction of the previously circulating Victoria lineage NP gene. These patterns were consistent with the reconstruction of the population genetic history for the NP gene where we observed additional peaks in genetic diversity following 2007/2008 when the Yamagata NP was acquired by Victoria viruses (*Figure 7*), indicating a major genome-level transition for Victoria lineage viruses. In contrast, the only inter-lineage reassortment events for the virus carrying the Yamagata HA occurred during 2002 and 2004 (red arrows in *Figure 8A,F*), when the NA and MP genes were derived from the Victoria lineage viruses, but these viruses went extinct within the same influenza season. In sum, these results show that the HA gene of Victoria viruses is placed in different genetic backgrounds at a higher rate and this is likely to have important fitness consequences.

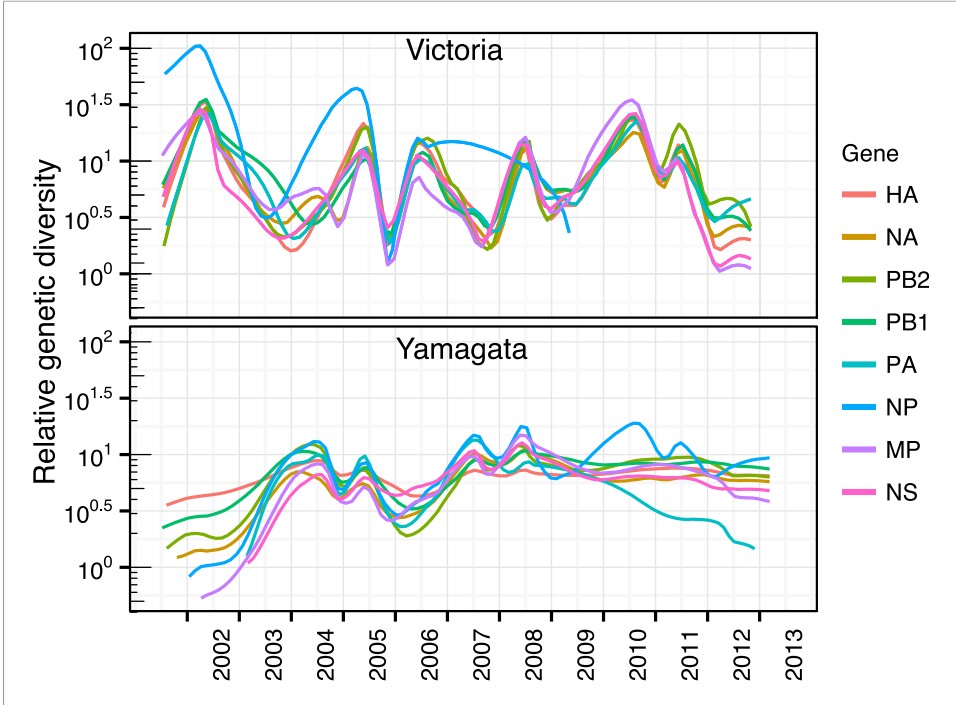

**Figure 7**. Genome wide evolutionary dynamics—relative genetic diversity. Relative genetic diversity of each gene segments of Victoria (black) and Yamagata (red) lineages estimated using the Gaussian Markov Random Fields (GMRF) Skyride model (as in **Figure 3**).

Phylogenies also suggest that the PB2 and PB1 gene trees (**Figure 8B,C**) exhibit deep divergence, similar to the HA gene where co-circulating viruses contain distinct Victoria and Yamagata genes. In contrast, the other gene segments exhibit relatively recent divergence indicating that the prevailing diversity of these genes originates from a single lineage. These results are consistent with a detailed investigation of long term reassortment patterns of influenza B virus lineages that revealed genetic linkage between the PB2, PB1 and HA protein genes (**Dudas et al., 2015**). Specifically, we observe that the PB2, PB1 and HA genes were consistently derived from a single lineage, except for the short-lived subpopulation in 2004.

## Differential selection pressure between lineages

Despite the marked differences in their epidemiological and evolutionary dynamics, the HA genes of the two influenza B lineages both evolved at a rate of approximately $2.0 \times 10^{-3}$ subs/site/year (**Table 1**), comparable to those previously estimated for a smaller ($n = 102$) global sample of influenza B viruses collected during 1989–2006 (**Chen and Holmes, 2008**) (mean nucleotide substitution rate of $2.15 \times 10^{-3}$ subs/site/year). These rates were considerably lower than those estimated for influenza A H3N2 and H1N1 viruses ($5.5 \times 10^{-3}$ subs/site/year, $4.0 \times 10^{-3}$ subs/site/year, respectively) (**Rambaut et al., 2008**). In contrast, analysis of the ratio of the number of nonsynonymous and synonymous substitutions per site ($d_N/d_S$) revealed significant differences between the influenza B virus lineages, with the Victoria lineage viruses having accumulated more nonsynonymous substitutions ($d_N/d_S = 0.19$) than the Yamagata lineage ($d_N/d_S = 0.13$) (p-value, <0.05). In addition, two amino acid residues in the Victoria HA (positions 212 and 214) were revealed to have experienced positive selection (p < 0.05), whereas no positively selected sites were observed in the Yamagata lineage over the time period studied. Similarly, the Victoria lineage exhibited a greater $d_N/d_S$ (ratio = 1.37) on internal vs external branches of the HA phylogeny compared to the Yamagata lineage (ratio = 0.98), indicating that amino acid changes have been fixed more frequently in Victoria than Yamagata lineage viruses (**Table 1**). Taken together, these results indicate that the Victoria lineage is under greater positive selection pressure, and hence likely to experience greater antigenic drift, than the more conserved Yamagata lineage.

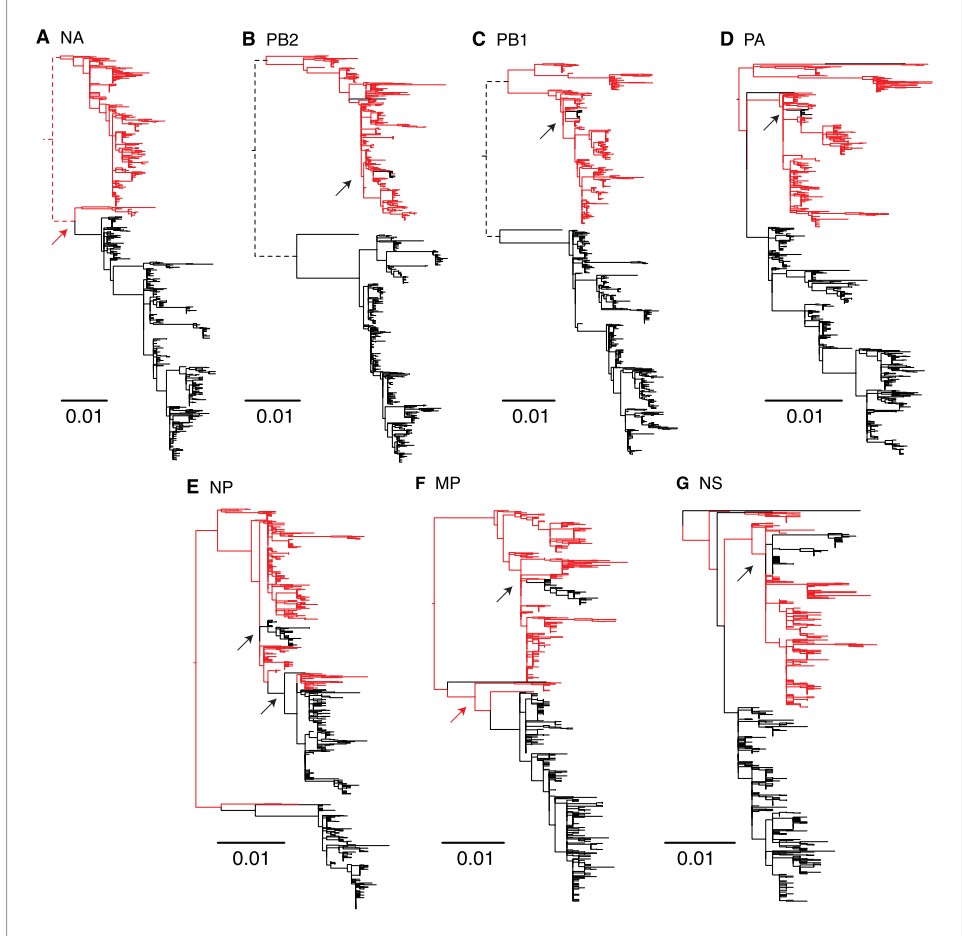

**Figure 8**. Genome wide evolutionary dynamics—reassortment. Evolutionary relationships of neuraminidase (**A**), polymerase basic 2 (**B**), polymerase basic 1 (**C**), polymerase acidic (**D**), nucleoprotein (**E**), matrix (**F**), and non-structural (**G**) genes of Victoria and Yamagata lineage viruses inferred using the maximum likelihood analysis of 908 full genome sequences. Lineages are coloured based on the HA lineage: Victoria (black) and Yamagata (red) and arrows highlight inter-lineage reassortment.

## Antigenic evolution

We constructed antigenic maps (*Smith et al., 2004*) using hemagglutination inhibition (HI) assay measurements for 87 Victoria and Yamagata viruses isolated during 2002–2013 and using 20 reference antigens and antisera (*Figure 9A*). These revealed that Victoria lineage viruses exhibited antigenic variation that generally clustered according to the year of isolation and phylogenetic distance, indicative of ongoing antigenic drift, and similar to that previously reported for H3N2 viruses (*Smith et al., 2004*; *Bedford et al., 2014*). In contrast, the antigenic distances for the Yamagata viruses had no correlation with time or phylogenetic distance and showed greater levels of antigenic cross-reactivity between antisera raised to both earlier and more recent viruses. Structural modeling showed that the degree of antigenic distance between strains of Victoria viruses was often linked to the proximity of single amino acid substitutions to the receptor binding pocket (RBP) of the HA (*Figure 9B*; see structural differences section below), in agreement with recent work on H3N2 (*Koel et al., 2013*). Importantly, the closer the amino acid change between two strains was to the RBP, the greater the antigenic difference between them.

## Heterogeneous age distributions of the lineages

In addition to genetic, antigenic, and evolutionary differences, we found a notable difference in the age distribution of infected cases for the two influenza B virus lineages (*Figure 10*) that was generally

**Table 1**. Nucleotide substitution rates (nucleotide substitutions/site/year) and selection pressures ($d_N/d_S$) of influenza B viruses from Australia and New Zealand during 2002–2013

| Segment* | Mean substitution rates (95% HPD) | Branch $d_N/d_S$ Global $d_N/d_S$ | Internal | External | Internal/External | Site $d_N/d_S$ No. +ve (sites) | No. −ve |
|---|---|---|---|---|---|---|---|
| **Victoria** | | | | | | | |
| PB2 | 1.49 (1.28–1.69) | 0.08 (0.07–0.09) | 0.02 | 0.03 | 0.55 | 0 | 373 |
| PB1 | 0.14 (0.12–0.16) | 0.08 (0.07–0.09) | 0.06 | 0.05 | 1.08 | 1 (474) | 402 |
| PA | 1.65 (1.44–1.88) | 0.13 (0.11–0.15) | 0.08 | 0.08 | 1.03 | 1 (700) | 334 |
| HA | 2.00 (1.74–2.57) | 0.19 (0.17–0.22) | 0.12 | 0.09 | 1.37 | 2 (212, 214) | 239 |
| NP | 1.04 (0.76–1.34) | 0.09 (0.07–0.12) | 0.07 | 0.05 | 1.22 | 0 | 49 |
| NA | 2.04 (1.72–2.36) | 0.31 (0.28–0.35) | 0.25 | 0.24 | 1.02 | 6 (46, 73, 106, 145, 146, 395) | 129 |
| MP | 1.44 (1.17–1.70) | 0.06 (0.04–0.09) | 0.00 | 0.02 | 0.01 | 0 | 87 |
| NS | 1.71 (1.38–2.06) | 0.45 (0.38–0.53) | 0.11 | 0.30 | 0.37 | 3 (116, 120, 249) | 13 |
| **Yamagata** | | | | | | | |
| PB2 | 2.00 (1.74–2.25) | 0.06 (0.05–0.07) | 0.03 | 0.02 | 1.44 | 0 | 443 |
| PB1 | 1.78 (1.56–2.00) | 0.07 (0.05–0.08) | 0.02 | 0.03 | 0.82 | 1 (357) | 392 |
| PA | 1.60 (1.35–1.84) | 0.10 (0.08–0.12) | 0.03 | 0.05 | 0.57 | 0 | 204 |
| HA | 2.01 (1.73–2.29) | 0.13 (0.11–0.16) | 0.07 | 0.07 | 0.98 | 0 | 245 |
| NP | 1.87 (1.65–2.10) | 0.10 (0.08–0.11) | 0.08 | 0.07 | 1.16 | 0 | 308 |
| NA | 2.25 (1.90–2.60) | 0.20 (0.17–0.24) | 0.30 | 0.18 | 1.70 | 1 (295) | 124 |
| MP | 2.20 (1.85–2.55) | 0.05 (0.03–0.07) | 0.05 | 0.02 | 2.08 | 0 | 102 |
| NS | 2.00 (1.66–2.39) | 0.33 (1.66–2.39) | 0.42 | 0.32 | 1.32 | 0 | 30 |

*Analysis was restricted to the non-overlapping regions of M1 and NS1, for the MP and NS segments, respectively.

consistent throughout our sampling period (*Figure 10—figure supplement 1*). On average, Victoria viruses infected a younger population (mean 16.8 years, median 11 years) compared to Yamagata viruses (mean 26.6 years, median 18 years). Although the proportion of cases under 6 years were similar in both lineages (28.8% of Victoria and 26.8% of Yamagata), there were 1.7 times more cases aged 6–17 years infected with a Victoria lineage virus (39.0% Victoria vs 22.7% Yamagata), while this ratio was almost reversed for those aged 18 years and over (32.2% Victoria vs 50.0% Yamagata; $\chi^2$, p < 0.0001) (*Table 2*). Thus, nearly 70% of Victoria lineage viruses were identified in children <18 years, whereas the Yamagata lineage exhibited a bimodal age distribution with a significant shift toward infections in individuals aged >25 years (*Figure 10*). These differences in age distribution are significant and unlikely to be explained by systematic bias because the same pattern was observed in both countries, and are consistent with data from Guangdong, China (*Tan et al., 2013*), and Slovenia (*Sočan et al., 2014*) during the 2009–2010 and 2010–2013 epidemic seasons, respectively.

A direct consequence of antigenic drift is the possibility for previously infected individuals to become reinfected. Subsequently, higher rates of antigenic drift in the Victoria lineage should lead to a more even age distribution of cases, whereas lower rates of antigenic drift should lead to an age distribution of cases that are skewed towards younger individuals. Although viruses of the Victoria lineage were consistently reported at a higher frequency during our surveillance period, the observed skew towards children runs counter to this expectation (*Figure 10*). One possible explanation is that the higher $R_e$ of the Victoria viruses reduces the mean age of infection, as expected in the case of a disease like influenza that imparts some immunity following infection (*Anderson and May, 1992*). Alternatively, the inability of Victoria viruses to infect an equivalent proportion of other age groups may mean that the relatively older population is better protected against this virus because of a broader immune response. The former scenario is supported by an increase in the mean age of infection from 15 years (median, 12) in 2008 to 20.5 years (median, 14) in 2011 for the B/Brisbane/60/

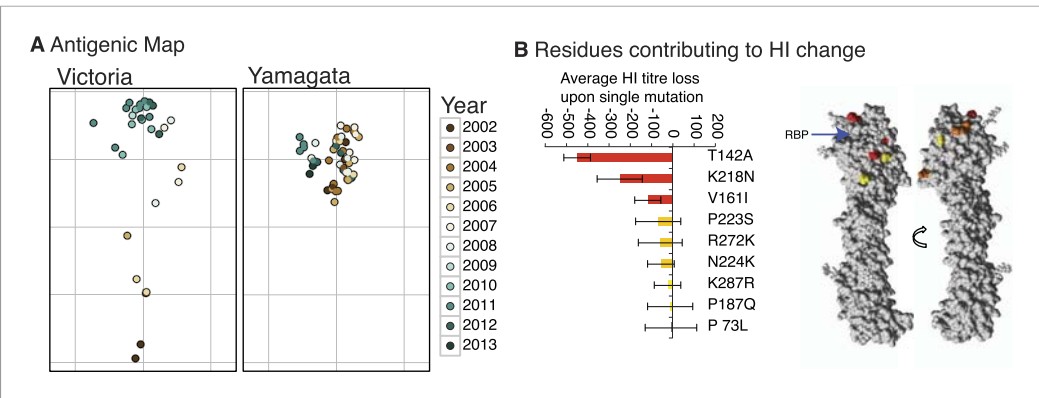

**Figure 9**. Antigenicity of influenza B viruses. Antigenic map showing relative antigenic differences of Victoria and Yamagata lineage viruses (circles) measured using the hemagglutinin inhibition (HI) assay for each strain and coloured by year of isolation (**A**). Residues contributing to HI titer changes (**B**). Among the nine amino acid changes that we detected between antigenically different Victoria viruses, three changes produced strong HI titer change (>100) (red), 3 medium (≈50) (orange) and 3 low (<20) (yellow). Changes that produced the strongest HI titer change were the closest to the receptor binding pocket (blue arrow), highlighting the significance of their proximity to HI titer change. Amino acids were mapped on previously resolved influenza B virus structure (PDB:4FQM). Detailed HI titer values and reference antigens used are provided in the Dryad source data (*Vijaykrishna et al., 2015*).

2008-like antigenic variant of the Victoria lineage, which coincided with a gradual drop in $R_e$ from its peak in 2008 (*Figure 5A*).

## Structural differences among influenza B viruses

Finally, we sought to determine whether differences in the evolutionary and epidemiological dynamics between the two influenza B lineages resulted from variation in HA structure and binding preferences. First, we compared amino acid substitutions per site within and between influenza virus lineages from 2002 to 2012 and mapped these onto structural models of representative influenza B virus strains (*Figure 11*). The higher rates of amino acid change observed in the Victoria HA (*Figure 11A*) were consistent with the stronger selective pressures on this viral lineage. Importantly, these changes occurred in three major clusters situated around 21, 29, and 37 Å to the RBP of the HA domain that

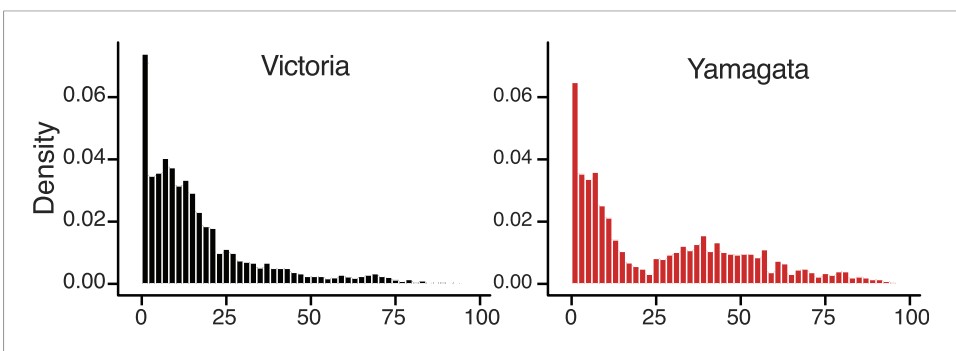

**Figure 10**. Age distribution of influenza B viruses. Density of age distribution of influenza B virus positive samples of Victoria (black) and Yamagata (red) lineages, collected from Australia and New Zealand during 2002–2013. Patient age was available for 5260 samples. The age distributions by lineage were compared by histogram using 2-year bins. Also see *Table 2* for comparison by age categories and Dryad source data for mean and median age for each year.
The following figure supplement is available for figure 10:

**Figure supplement 1**. Year-wise age distribution of influenza B viruses.

**Table 2**. Age distribution by group

| | Victoria | | Yamagata | | |
|---|---|---|---|---|---|
| Age | n | % | n | % | p value* |
| <6 | 1007 | 28.8 | 473 | 26.8 | |
| 6–17 | 1361 | 39 | 402 | 22.7 | |
| ≥18 | 1124 | 32.2 | 893 | 50.5 | |
| Total | 3492 | 100 | 1768 | 100 | <0.0001 |

*Age categories were compared by lineage using a $\chi^2$ test.

also comprises potential antigenic sites. Notably, all changes in the closest cluster (21 Å) were comprised exclusively of Victoria lineage amino acid changes, while the few changes observed in Yamagata lineage viruses were distant to the RBP (*Figure 11C*). Overall, however, amino acid changes in both influenza B virus lineages were less frequent than those in influenza A viruses sampled over a similar time period, with the H3N2 viruses showing more extensive structural change (*Figure 11—figure supplement 1*).

Notably, we also observed fundamental structural differences between the lineages (*Figure 11B*). Crystal structures showed extensive backbone differences around amino acid sites 165 and 180 that lie near the RBP as well as residue differences in the helix close to where $\alpha$-2,3 and $\alpha$-2,6 ligands differ structurally, thereby potentially influencing receptor binding (*Figure 11D*). Previous experiments suggest that Yamagata viruses bind predominantly to $\alpha$-2,6-linked sialic acid host receptors while Victoria viruses have both $\alpha$-2,3 and $\alpha$-2,6 binding capacities (*Wang et al., 2012*; *Velkov, 2013*). Binding differences may also originate in part from differences in *N*-glycosylation patterns between the lineages (*Figure 11E, 12*). While both lineages share a possible glycan at Asn 160, only Victoria has a functional *N*-glycosylation site at Asn 248, although its distance from the receptor may account for only a limited role in binding differences. On the other hand, *N*-glycosylation at Asn 212 occurs in both lineages but has a lower overall frequency in Victoria strains. In light of the positive selection acting on codon sites 212 and 214 in the Victoria lineage, it is interesting to note that amino acid changes in either site would abolish the *N*-glycosylation at 212, thereby highlighting a possible functional consequence of gain or loss of a glycan at this site. Furthermore, position 212 is located at the exit of the RBP which is used by $\alpha$-2,3-linked sialic acid host receptors, and loss of *N*-glycosylation at 212 consequently adds capacity to bind $\alpha$-2,3 and not just $\alpha$-2,6-linked sialic acid host receptors (*Figure 11E*). Importantly, all our sequenced viruses have been passaged in MDCK cells to avoid egg adaptation artifacts in this context (*Gambaryan et al., 1999*). Interestingly, we observed that loss of *N*-glycosylation at site 212 was associated with an increased proportion in the younger (0–5 years) age group (*Figure 12*). We therefore hypothesize that subtle differences in the prevalence of $\alpha$-2,3- and $\alpha$-2,6-linked glycans on the cells of the respiratory tract of young children compared to adults (*Nicholls et al., 2007*; *Walther et al., 2013*), combined with partial changes in glycosylation patterns, could account for the observed differential age distribution of the two influenza B lineages.

## Conclusions

The genomic and epidemiological data analyzed here provide important insights into the phylodynamics of the two lineages of influenza B virus currently circulating in humans. In particular, we find significant differences in the evolutionary and epidemiological dynamics between the Victoria and Yamagata lineages (*Table 3*). Central to this is the observation that the phylodynamic pattern of the Victoria lineage HA gene is indicative of a virus population under greater selection pressure that escapes host immunity by accruing beneficial amino acid substitutions in the HA gene. Indeed, theory predicts that the highest rate of viral adaptation occurs at intermediate levels of immune pressure (*Grenfell et al., 2004*) which may characterize the Victoria lineage. Such an evolutionary pattern ensures that there is a constant supply of susceptible individuals for Victoria lineage viruses—both naïve and reinfected individuals which in turn increases $R_e$—which then exhibit a pattern of genomic diversity and lineage turnover that is significantly faster and more periodic than Yamagata lineage viruses.

In contrast, the phylodynamic patterns exhibited by Yamagata viruses are indicative of a virus population that exhibits slower and less periodic dynamics, reflected in a lower and more consistent $R_e$, in turn suggesting that these viruses are under weaker immune selection pressure and accordingly experience weaker antigenic drift. Interestingly, clinical trials of influenza B virus vaccination in children (*Skowronski et al., 2011*) and experimental infection of mice (*Skowronski et al., 2012*) showed that the Yamagata antigens produced a stronger immune response than the Victoria antigens. If natural

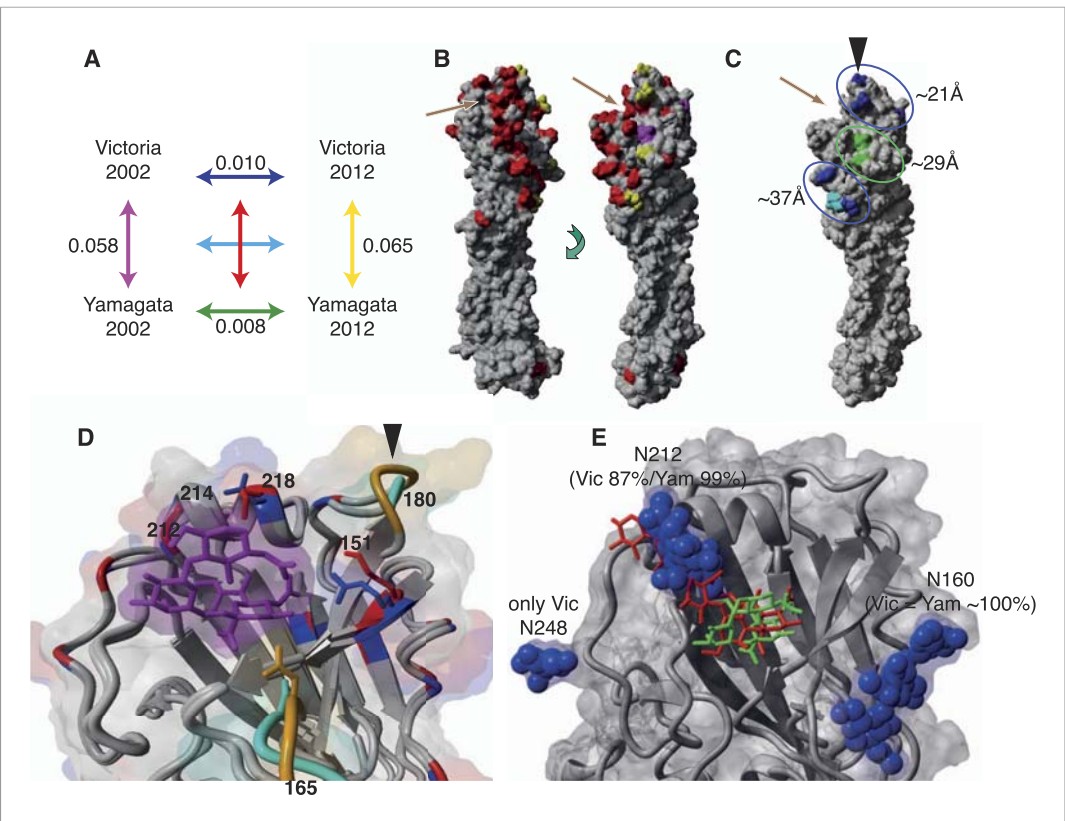

**Figure 11**. Structural view of the HA showing mutational accumulation and lineage differences. Amino acid changes observed within and between influenza B virus lineages (**A**). Arrow colours in (**A**) correspond to inter- (**B**) or intra- (**C**) lineage amino acid changes, based on previously resolved crystal structure (PDB:4FQM). Amino acids in red represent differences between the two lineages that were retained over all sampling years; yellow represents differences that are newly observed in 2012 compared to 2002; and magenta represents changes lost in 2012 compared to 2002. Amino acids in blue and green represent changes that occurred in Victoria and Yamagata viruses between 2002 and 2012, respectively; whereas cyan represents difference between 2002 and 2012 shared between both lineages. These amino acid changes occur in regions that cluster around 21, 29, and 37 Å distant from the RBP (**C**). Structural differences in RBP among recent Victoria (B/Brisbane/60/2008) and Yamagata (B/Florida/4/2006) strains with a human-like $\alpha$-2,6 host receptor analogue (magenta) modeled within the viral RBP (**D**). D was based on crystal structures PDB:4FQM and PDB: 4FQJ with side-chains minimized after addition of ligand from PDB:2RFU through superposition. Regions differing in backbone conformation are shown in orange for Victoria and cyan for Yamagata, while conserved regions are shown in gray. Residues with conserved backbone structure but different amino acid side-chains are shown in red for Victoria and blue for Yamagata. Side-chains are shown only for residues within 5 Å of the receptor ligand and differing between the lineages. Structural view of receptor binding pocket with $\alpha$-2,6- (green) and $\alpha$-2,3-linked (red) host receptor and glycans (blue) (**E**). E was based on crystal structure PDB:4FQM, with the addition of ligands from PDB:2RFU and PDB:2RFT through superposition and no minimization. The presence of a glycan on site 212 allows binding only to 2,6-linked receptors, while loss of the glycan allows binding to both $\alpha$-2,3- and $\alpha$-2,6-linked receptors. Brown arrows (**B** and **C**) indicate relative position of receptor binding pocket (RBP), whereas black arrow heads (**C** and **D**) point to site of known antigenic cluster transition (**Koel et al., 2013**).

The following figure supplement is available for figure 11:

**Figure supplement 1**. Structural view of mutational drift in influenza A and B viruses.

infection with influenza B virus was similar, this would imply that Yamagata viruses are less able to evolve through antigenic drift and therefore escape the immune response (*Grenfell et al., 2004*).

We propose that these fundamental differences in evolutionary and epidemiological dynamics are driven by differences in hemagglutinin binding preferences. Specifically, Victoria viruses appear to have both $\alpha$-2,3- and $\alpha$-2,6-linked sialic acid binding capacities (*Wang et al., 2012*; *Velkov, 2013*),

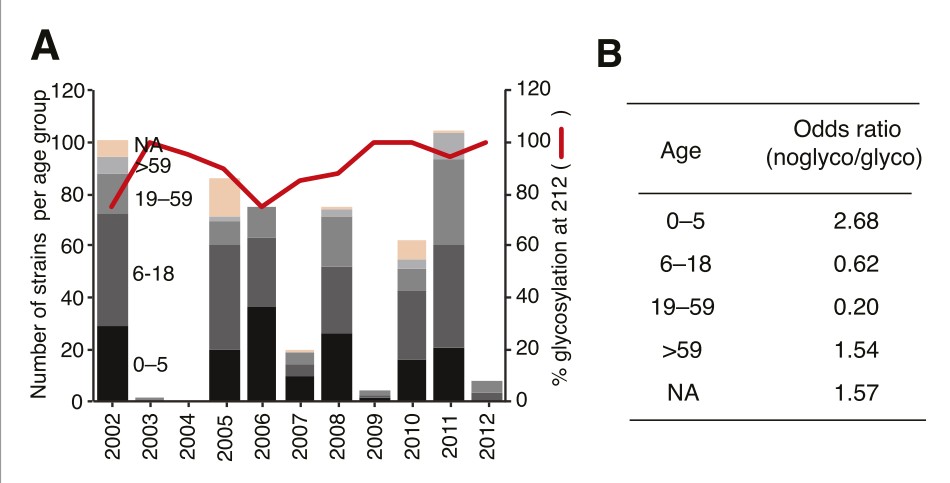

**Figure 12**. Glycosylation at Asn 212 and correlation with age groups for Victoria viruses. Yamagata viruses showed five instances of glycosylation loss at 212, compared to 71 instances in Victoria, hence Victoria lineage strains have been analyzed in detail here. Temporal distribution of age groups and glycosylation at 212 for all Victoria strains (**A**). Summary of odds ratio (OR) for association of glycosylation loss at 212 with the different age groups (**B**). OR values >1 indicate that it is more likely to find a 212 loss in the respective age group, whereas values <1 indicate that 212 losses are less likely to be found in the respective groups. The following guideline helps judging significance of OR: strong positive association >3; moderate positive association 1.5–3; moderate negative association 0.33–0.66; strong negative association <0.33.

while Yamagata viruses predominantly bind to $\alpha$-2,6-linked glycans on cells in the human respiratory tract. Experimental studies in children (aged up to 7) (*Nicholls et al., 2007*) and adults have shown that the respiratory tissue of children mainly have $\alpha$-2,3-linked receptors with a lower level of $\alpha$-2,6-linked receptors than adults, and these differences among the different age groups may in part account for the different age distribution of the two B lineages. In turn, the greater propensity to infect children will increase $R_e$, initiating the epidemiological and evolutionary pattern that characterizes the Victoria lineage. It remains to be determined whether the broadly equivalent phylodynamic differences between the H3N2 and seasonal H1N1 types of influenza A virus are similarly due to basic differences in the structure of their respective HA proteins. Furthermore, to better understand the bimodal age distribution in Yamagata, where a significant reduction of infection was observed among the older children–young adult group (<25 years), additional experimental studies of the receptor distribution in all age groups are necessary.

**Table 3**. Summary of evolutionary and epidemiological characteristics of influenza B virus lineages

| Characteristics | Victoria | Yamagata |
|---|---|---|
| Age distribution | younger (mean 16.8, median 11) | older (mean 26.6, median 18) |
| Genetic diversity | strong seasonal changes | weak seasonal changes |
| R (medians) | higher (1.13–1.27) | lower (1.08–1.14) |
| Positive selection | stronger | weaker |
| Antigenic drift | relatively strong | relatively weak |
| Reassortment | high inter-sublineage reassortment, with lower intra-sublineage reassortment | low inter-sublineage reassortment, with greater intra-sublineage reassortment |
| Receptor binding preference | $\alpha$-2,3- and $\alpha$-2,6-linked sialic acid | mainly $\alpha$-2,6 linked sialic acid |

These observations have implications for the future control of influenza B virus in the human population. While the co-circulation of divergent Yamagata viruses reported here has and can confound the accurate selection of vaccine strains, our analyses also indicate that the Yamagata viruses are under weaker positive selection and antigenic drift, and, on average, infect an older group of people who are more likely to have a higher level of cross-reactive antibodies to the B lineage viruses compared to children. As a consequence, there is a greater chance that, given sufficient coverage, Yamagata viruses might experience a major drop in prevalence over time through targeted control methods, such as the extensive use of quadrivalent influenza vaccines containing both B lineages, in contrast to the more adaptable Victoria viruses.

## Materials and methods

### Surveillance

Influenza B positive samples collected between 2002 and 2013 from subjects in eastern Australia (Victoria, New South Wales and Queensland) and from New Zealand and associated metadata, including date of isolation and age of host, were sent to the WHO Collaborating Centre for Reference and Research on Influenza, Melbourne, from National Influenza Centres and other laboratories as part of the World Health Organization Global Influenza Surveillance and Response System (WHO GISRS). Data deposited in Dryad data repository under DOI: 10.5061/dryad.n940b (*Vijaykrishna et al., 2015*).

### Virus isolation

Influenza B viruses were isolated or re-isolated in MDCK cells (ATCC-CCL 34) from original clinical samples or virus isolates and typed as B/Yamagata or B/Victoria using HI analysis or by molecular assay (*Deng et al., 2013*). Viruses were stored at −80°C until sequenced.

### Sequencing of viral RNA genome

We sequenced the complete genomes of 908 laboratxory confirmed influenza B virus MDCK or MDCK-SIAT cell propagated isolates passaged 1–4 times from eastern Australia and New Zealand using a novel methodology (*Zhou et al., 2014*). Influenza B virus genomes were amplified using the universal influenza B genomic amplification strategy that enables amplification of the complete genome of any influenza B virus in a one-step single tube/well reaction. Specifically, RNA was isolated from 130 µl of culture supernatant using ZR-96 Viral RNA Kit (Zymo Research, Irvine, CA) and eluted in 30 µl of RNase-free water. 3 µl of the RNA was mixed with FluB Universal Primer Cocktail (*Zhou et al., 2014*) and converted to cDNA and amplified with the SuperScript III One-Step RT-PCR System (Life Technologies, Grand Island, NY). The amplicons were fragmented, flanked by sequencing adaptors, clonally amplified onto IonSphere particles, and sequenced on the Ion Torrent PGM platform following manufacturer's instruction. The sequence reads were sorted by bar code to separate different viruses and used to assemble viral genomes (sequence accession numbers are available in the Dryad data repository under DOI: 10.5061/dryad.n940b).

### Phylogenetic analysis

Sequences were curated, and maximum likelihood (ML) phylogenetic trees were inferred for each gene segment independently from the samples described above. ML trees were estimated using iqtree v0.9.5 (*Minh et al., 2013*) using the best-fit nucleotide substitution model, chosen by the Bayesian Information Criterion (BIC). The data were further divided into separate lineages (i.e., Victoria and Yamagata) and time-scaled phylogenies and rates of nucleotide substitution for each were inferred using a relaxed molecular clock model in a Bayesian Markov Chain Monte Carlo (MCMC) framework with the program BEASTv1.8 (*Drummond et al., 2012*) that incorporates virus sampling dates to concurrently estimate phylogenetic trees, rates of nucleotide substitution, and the dynamics of population genetic diversity using a coalescent based approach. The analysis was conducted with a General Time Reversible (GTR) model with a gamma ($\Gamma$) distribution of among-site rate variation and a time-aware linear Bayesian skyride coalescent tree prior (*Minin et al., 2008*). We performed at least two independent analyses per data set for 100 million generations sampled every 10,000 runs. After the appropriate removal of burn-in (10–20% of samples in most cases), a summary Maximum Clade Credibility (MCC) tree was inferred and visualized with Figtree v1.4 (*Rambaut, 2014*). Support for individual nodes is reflected in posterior probability values, and statistical uncertainty is given by 95% Highest Posterior Density (HPD) intervals.

The MCC trees were also used to estimate the genealogical pairwise diversity by averaging the time distance between contemporaneous sample pairs with a 1 month window (*Zinder et al., 2013*).

The past population dynamics of each linage were compared using a Bayesian skyride analysis in BEAST, which utilizes a Gaussian Markov Random Field (GMRF) smoothing prior to estimate the changes in relative genetic diversity in successive coalescent intervals (*Minin et al., 2008*). In the absence of natural selection (i.e., under a strictly neutral evolutionary process), the genetic diversity measure obtained reflects the change in effective number of infections over time ($N_{e}t$, where $t$ is the average generation time). However, because natural selection can play a major role in the evolution of the influenza HA, these are interpreted as 'relative genetic diversity', and which is consistent with previous studies of influenza A virus (*Rambaut et al., 2008*). Sequence alignments with input parameters are available under Dryad data repository under DOI: 10.5061/dryad.n940b.

## Phylogeography and migration rate estimates

We used a continuous-time Markov chain (CTMC) phylogeographic process (*Minin and Suchard, 2008*; *Lemey et al., 2009*) to estimate counts of migration to and from Australia and New Zealand, similar to previous studies (*Nunes et al., 2012*; *Bahl et al., 2013*). Briefly, global influenza B virus HA sequences and their associated spatial locations and isolation dates were downloaded from GenBank for the years for which we estimated an effective reproductive number in the phylodynamic analysis (see below). Spatial locations of the isolates were transformed to represent two discrete states: the region of interest (Australia and New Zealand) and the rest of the world. Phylogeographic events were estimated independently for each of the identified years using an asymmetric CTMC process (*Minin and Suchard, 2008*), with the estimated state transition counts (import and export) between the two discrete states estimated using a Markov Jump count approach. This phylogeographic inference was implemented in BEAST 1.8 (*Drummond et al., 2012*) similar to the temporal phylogenies described above. The resulting log files were used in extracting the net migration counts and mean non-zero transition rates.

## Phylodynamic analysis

To estimate epidemiological parameters (specifically the effective reproductive number, $R_e$) for each epidemic of virus lineages in Australia and New Zealand, we used the birth–death susceptible-infected-removed (BDSIR) model (*Kühnert et al., 2014*). The BDSIR analysis was also conducted with a GTR + Γ substitution model, with epidemiological dynamics estimated jointly with the phylogenies for each virus lineage. The model assumes a closed SIR epidemic in each season for the underlying host population. The initial number of susceptible individuals $S_0$ could not be estimated and was therefore initially fixed to 4,000,000 (results reported in the main text). Analysis under different $S_0$ values, ranging from 40,000 to 10 million, showed that the estimates of reproductive numbers ($R_e$) are robust to the choice of $S_0$. The BDSIR analyses utilized $m = 100$ intervals for the approximation of the SIR dynamics. Incidence and prevalence were computed from the posterior distributions of the SIR trajectories, and the relevant plots show their median values.

## Molecular adaptation

Selection pressures for each gene segment, lineage, and individual codon were estimated as the ratio of the number of nonsynonymous substitutions per nonsynonymous site ($d_N$) to the number of synonymous substitutions per synonymous site ($d_S$). Estimates were obtained using the Single Likelihood Ancestor Counting (SLAC) (*Kosakovsky Pond and Frost, 2005*) and Fast Unconstrained Bayesian AppRoximation (FUBAR) (*Murrell et al., 2013*) methods, accessed through the Datamonkey webserver of the HyPhy package (*Delport et al., 2010*). In addition, the $d_N/d_S$ ratio for the internal and external branches of the Victoria and Yamagata HA phylogenies was estimated separately using the CODEML program (two-ratio model) available in the PAML suite (*Yang, 2007*).

## HI assay and antigenic cartography

Representative viruses from each lineage were sub-sampled and tested for antigenic reactivity by a hemagglutination inhibition (HI) assay using a panel of reference ferret antisera that were available for each influenza B lineage (raw HI titers are available in the Dryad data repository under DOI: 10.5061/dryad.n940b) and the subsequent antigenic profile was used to generate antigenic maps

(*Cai et al., 2010*) for each lineage. HI assays were performed as described previously (*WHO Global Influenza Surveillance Network, 2011*) using panels of post-infection ferret sera raised against representative viruses from both B/Victoria lineage or the B/Yamagata lineage collected from 2000 to 2013. Turkey red blood cells were used to detect unbound virus and the HI titer was determined as the reciprocal of the last dilution that contained non-agglutinated RBC. Normalized titers from the HI assay were compiled for antigenic cartography analysis. The HI matrix was used in a multi-dimensional scaling (MDS) plot algorithm to chart the antigenic distances between isolates tested in a two-dimensional map (*Cai et al., 2010*), through the AntigenMap webserver (*Wan, 2010*). To identify residues contributing most to HI titer changes, pairwise comparison of sequences with a single amino acid difference were conducted.

## Computational structural modeling

Finally, sequence data of the HA segment from each lineage were used to construct structural models (*Krieger et al., 2009*; *Webb and Sali, 2014*). To identify those residues that contribute most to antigenic drift in Victoria viruses, we compared the HA amino acid sequences of all pairs of HI assay tested strains using the Smith-Waterman algorithm. If only a single mutation difference was found, we calculated the respective average HI titer change for occurrences of this mutation. These amino acid sites were then mapped on the crystal structure PDB:4FQM (*Dreyfus et al., 2012*) and visualized using YASARA (*Krieger et al., 2009*).

Amino acid substitutions per site between pairs of HA sequences were calculated using MEGA5 (*Tamura et al., 2011*) under the Jones-Taylor-Thornton (JTT) amino acid substitution model. We constructed structural models using MODELLER (*Webb and Sali, 2014*) (five models each with and without ligand, best model selected by DOPE quality score), structural alignments were conducted using MUSTANG (*Konagurthu et al., 2006*) and visualized using YASARA (*Krieger et al., 2009*). To identify structural changes occurring on the HA proteins of influenza A subtypes and influenza B virus lineages over a 10-year period, we selected the HA protein sequences of the following virus strains: influenza B Victoria lineage, B/Sydney/1/2002 and B/Sydney/205/2012; Yamagata lineage, B/Victoria/341/2002 and B/Victoria/831/2012; influenza A H1N1 virus, A/Brisbane/59/2007 and A/Malaysia/11641/1997 and influenza A H3N2 virus, A/Perth/16/2009 and A/Moscow/10/1999. Crystal structure templates used for computational modeling include PDB:4FQM (*Dreyfus et al., 2012*) (influenza B virus), PDB:3UBE (*Xu et al., 2012*) (H1N1), and PDB:2YP4 (*Lin et al., 2012*) (H3N2).

Differences in the receptor binding pocket region of the two influenza B lineages were visualized using B/Brisbane/60/2008 (PDB:4FQM [*Dreyfus et al., 2012*]) and B/Florida/4/2006 (PDB:4FQJ [*Dreyfus et al., 2012*]) with the addition of an $\alpha$-2,6-linked host receptor analogue ligand from a known complex (PDB:2RFU [*Wang et al., 2007*]) and targeted side-chain minimization of residues within 8 Å of the ligand through short simulated annealing molecular dynamic simulations in YASARA (*Krieger et al., 2009*) as previously benchmarked to ensure realistic results.

We also used YASARA (*Krieger et al., 2009*) to visualize the role of glycosylation on Asn at position 212 for $\alpha$-2,3- vs $\alpha$-2,6-linked host receptor ligands by schematically superimposing both ligands (PDB:2RFT [*Wang et al., 2007*] and PDB:2RFU [*Wang et al., 2007*]) into their respective positions within the receptor binding pocket of a fully glycosylated influenza B HA head (PDB:4FQM [*Dreyfus et al., 2012*]).

## Acknowledgements

The authors thank Tasoula Mastorakos for assistance in sample preparation and shipping, Malet Aban for HI assays and helpful discussions with Professor Heath Kelly, VIDRL. We also thank the Australian National Notifiable Diseases Surveillance Systems (NNDSS) for provision of data. Several additional laboratories kindly provided viruses used in this research and the authors would like to acknowledge these: Margaret C Croxson and staff at Clinical HOD, Virology/Immunology, LabPlus, Auckland City Hospital, Auckland, NZ; Julian Druce and staff from Victorian Infectious Diseases Reference Laboratory, North Melbourne, Victoria, Australia; Noelene Wilson and staff at Pathology North, NSW Health, Newcastle, NSW, Australia; Bruce Harrower and staff from Public and Environmental Health Virology, Forensic and Scientific Services, Queensland Health, Coopers Plains, Queensland, Australia. The authors thank Asmik Akopov, Amy Ransier, and Michael Mohan for their technical assistance in next-generation sequencing library construction, Dan Katzel for sequence database engineering and management, and Dana Busam for next-generation sequencing.

# Additional information

## Funding

| Funder | Grant reference number | Author |
| --- | --- | --- |
| National Institutes of Health (NIH) | Contracts HHSN266200700005C, HHSN272200900007C and HHSN272201400006C and R01 GM080533 | Dhanasekaran Vijaykrishna, Edward C Holmes, Udayan Joseph, Mathieu Fourment, Yvonne CF Su, Rebecca Halpin, Raphael TC Lee, Yi-Mo Deng, Vithiagaran Gunalan, Xudong Lin, Timothy B Stockwell, Nadia B Fedorova, Bin Zhou, Natalie Spirason, Denise Kühnert, Tanja Stadler, Anna-Maria Costa, Dominic E Dwyer, Q Sue Huang, Lance C Jennings, William Rawlinson, Sheena G Sullivan, Aeron C Hurt, Sebastian Maurer-Stroh, Gavin JD Smith, Ian G Barr |
| Department of Health and Ageing, Australian Government | WHO Centre funding | Dhanasekaran Vijaykrishna, Yi-Mo Deng, Natalie Spirason, Sheena G Sullivan, Aeron C Hurt, Gavin JD Smith, Ian G Barr |
| Agency for Science, Technology and Research (A*STAR) | Duke-NUS Signature Research Program | Dhanasekaran Vijaykrishna, Udayan Joseph, Yvonne CF Su, Gavin JD Smith |
| Ministry of Health -Singapore (MOH) | Duke-NUS Signature Research Program | Dhanasekaran Vijaykrishna, Udayan Joseph, Yvonne CF Su, Ian G Barr |
| Ministry of Education - Singapore (MOE) | Academic Research Fund grant MOE2011-T2-2-049 | Dhanasekaran Vijaykrishna |
| National Health and Medical Research Council (NHMRC) | NHMRC Australia Fellowship | Edward C Holmes |
| Swiss National Science Foundation (Schweizerische Nationalfonds) | | Denise Kühnert, Tanja Stadler |
| Agency for Science, Technology and Research (A*STAR) | 12/1/06/24/5793 | Aeron C Hurt, Sebastian Maurer-Stroh |
| National Health and Medical Research Council (NHMRC) | 12/1/06/24/5793 | Aeron C Hurt, Sebastian Maurer-Stroh |

The funders had no role in study design, data collection and interpretation, or the decision to submit the work for publication.

## Author contributions

DV, ECH, Conception and design, Acquisition of data, Analysis and interpretation of data, Drafting or revising the article; UJ, Y-MD, ACH, DEW, Acquisition of data, Analysis and interpretation of data; MF, Phylogenetic analysis and interpretation; YCFS, Genetic data curation, Phylogenetic analysis and interpretation; RH, Oversaw logistical and technical aspects of the viral sequencing; RTCL, VG, Structural data analysis and interpretation; XL, Viral genome purifications and amplification; TBS, Directed viral sequence assembly and informatics; NBF, Viral genome sequence finishing and closure; BZ, Viral genome sequencing technical oversight; NS, A-MC, DED, QSH, LCJ, WR, Collected and

curated virus samples and associated metadata; DK, TS, SM-S, Analysis and interpretation of data, Drafting or revising the article; VB, Phylodynamic analysis and interpretation; SGS, Statistical and epidemiological analysis, Acquisition of data, Analysis and interpretation of data; GJDS, Conception and design, Acquisition of data, Analysis and interpretation of data, Drafting or revising the article; IGB, Conception and design, Acquisition of data, Analysis and interpretation of data, Drafting or revising the article, Contributed unpublished essential data or reagents

# Additional files

## Major datasets

The following dataset was generated:

| Author(s) | Year | Dataset title | Dataset ID and/or URL | Database, license, and accessibility information |
|---|---|---|---|---|
| Vijaykrishna D, Holmes EC, Joseph U, Fourment M, Su YCF, Halpin R, Lee RTC, Deng Y-M, Gunalan V, Lin X, Stockwell TB, Fedorova NB, Zhou B, Spirason N, Kühnert D, Bošková V, Stadler T, Costa A-M, Dwyer DE, Huang QS, Jennings LC, Rawlinson W, Sullivan SG, Hurt AC, Maurer-Stroh S, Wentworth DE, Smith GJD, Barr IG | 2014 | Data from: The contrasting phylodynamics of human influenza B viruses | doi:10.5061/dryad.n940b | Available at Dryad Digital Repository under a CC0 1.0 Public Domain Dedication. |

The following previously published datasets were used:

| Author(s) | Year | Dataset title | Dataset ID and/or URL | Database, license, and accessibility information |
|---|---|---|---|---|
| Dreyfus C, Laursen NS, Wilson IA | 2012 | Structure of B/Brisbane/60/2008 Influenza Hemagglutinin | http://www.rcsb.org/pdb/explore.do?structureId=4FQM | Publicly available at RCSB Protein Data Bank. |
| Xu R, Wilson IA | 2012 | Influenza hemagglutinin from the 2009 pandemic in complex with ligand LSTc | http://www.rcsb.org/pdb/explore.do?structureId=3ube | Publicly available at RCSB Protein Data Bank. |
| Xiong X, Lin YP, Wharton SA, Martin SR, Coombs PJ, Vachieri SG, Christodoulou E, Walker PA, Liu J, Skehel JJ, Gamblin SJ, Hay AJ, Daniels RS, McCauley JW | 2012 | Haemagglutinin of 2004 Human H3N2 Virus in Complex with Human Receptor Analogue LSTc | http://www.rcsb.org/pdb/explore.do?structureId=2YP4 | Publicly available at RCSB Protein Data Bank. |
| Dreyfus C, Laursen NS, Wilson IA | 2012 | Influenza B/Florida/4/2006 hemagglutinin Fab CR8071 complex | http://www.rcsb.org/pdb/explore.do?structureId=4FQJ | Publicly available at RCSB Protein Data Bank. |
| Wang Q, Tian X, Chen X, Ma J | 2007 | Crystal structure of influenza B virus hemagglutinin in complex with LSTc receptor analog | http://www.rcsb.org/pdb/explore.do?structureId=2RFU | Publicly available at RCSB Protein Data Bank. |
| Wang Q, Tian X, Chen X, Ma J | 2007 | Crystal structure of influenza B virus hemagglutinin in complex with LSTa receptor analog | http://www.rcsb.org/pdb/explore.do?structureId=2rft | Publicly available at RCSB Protein Data Bank. |

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
