## [Decision Letter]

Thank you for sending your work entitled “The contrasting phylodynamics of human
influenza B viruses” for consideration at *eLife*. Your article
has been favorably evaluated by Diethard Tautz (Senior editor) and 2 reviewers, one of
whom is a member of our Board of Reviewing Editors.

The Reviewing editor and the other reviewer discussed their comments before we reached
this decision, and the Reviewing editor has assembled the following comments to help you
prepare a revised submission.

We agreed that your paper provides the first comprehensive portrait of influenza B
phylodynamics and epidemiology and that the comparisons of the Yam and Vic lineages with
each other and with the dominant seasonal influenza A lineages greatly advance our
understanding of influenza dynamics in general. We did, however, have a number of
concerns regarding some of the presented analysis and the presentation of some of the
results.

1) Reassortment analysis: we were not convinced that the comparison of reassortment
between different seasonal influenza strains via multidimensional scaling is
appropriate. The analysis of A/H3N2 (Rambaut et al., 2008) uses a different distance
measure and it is not clear whether the results are comparable. It is also not clear how
sensitive and robust MDS based on the proposed distance measure is. It may be hard to
conclude quantitative differences from this analysis as the dispersion in the plot will
not only reflect the number of reassortment events, but also whether those events
shuffle closely or distantly related lineages. In terms of inter-lineage reassortment,
the MDS in Figure 10 does not add much to Figure 9 (which shows each probable reassortment
event) and meaningful quantification of intra-lineage reassortment would require going
beyond the MDS. We suggest cutting the MDS. Also, results by Dudas, Bedford, Lycett,
Rambaut, MBE, 2014 should be discussed; the preprint has been available since March.

2) More direct measures of genetic diversity: Figure 3 and Figure 8 are labeled as
“relative genetic diversity” (relative to what?). We would like to see a
more direct measure of diversity such as the average pairwise distance in windows of a
few months. We would also like to ask you to offer a clearer interpretation of the GMRF
results. The GMRF method gives smoothed estimate of the rate of coalescence and is based
on a neutral coalescent model. This measure is not the same as genetic diversity and the
meaning of estimates based on a neutral coalescent is not clear.

The take home message from these plots seems to be that Vic has a
“burstier” and shallower phylogeny than Yam, with fewer lineages carrying
over from one season to another. This would show up as spikes and troughs in the rate of
coalescence and is consistent with the higher R0 and the smaller number of introductions
for Vic. If this is the desired interpretation, please discuss as such.

3) Phylodynamic analysis: how were the reported years chosen? (please double check the
color code in Figure 2. 2011 seems to be fully dominated by Yam, but you report Vic
estimates in Figures 4 and 5). The discussion of
the implications of higher R0 for Vic needs clarification. Prevalence within one season
depends on the start of the exponential growth phase, R0, and the number of independent
introductions (i.e., the number of lineages at the beginning if the growth phase). The
two strains differ in the latter two of these characteristics in ways that affect
prevalence in opposite ways. This needs more careful discussion: the assertion that
“these results are consistent with the more frequent detection of Victoria
lineage viruses during our sampling period (see Source data in Dryad), indicating that
Victoria viruses infected a higher proportion of the population” (in the Results
and discussion section) is a tautology (more frequent detection corresponding to higher
prevalence…). Would it be possible to explicitly show that exponential growth of
Vic and Yam differs between years? The >5000 samples used for the age
distribution should be sufficient to directly show differences in dynamics if date
information beyond year is available. If this is not possible, Figure 6 should be presented on a log scale and real incidence data
should be included.

4) Age distribution and glycosylation: Figure 16C shows odds ratios of for an
association between glycosylation at 212 and infections in age group 0-5. In Figure 16B,
however, it would be much more useful to show that the fraction without 212
glycosylation correlates with the fraction of in age group 0-5. Otherwise, the
correlation is confounded by the total number of strain. When discussing the structural
implications of the observed sequences changes, more care should be taken to distinguish
hard evidence from previously solved structures (give PDB IDs and references in figure
or caption!), modeling, and hypothesis. Figure 15 could be absorbed as a supplement to
Figure 14. The “three major clusters” in Figure 14C are not as clear as
the text suggests. Elaborate or tone down.

5) Presentation and shortening: the manuscript is long with 16 figures in the main text
and its length dilutes the most important points. Also, please make every effort to
include parameters of programs used, scripts, and input files (such as BEAST xmls).

---

## [Author Response]

*1) Reassortment analysis: we were not convinced that the comparison of
reassortment between different seasonal influenza strains via multidimensional
scaling is appropriate. The analysis of A/H3N2 (Rambaut et al., 2008) uses a
different distance measure and it is not clear whether the results are comparable. It
is also not clear how sensitive and robust MDS based on the proposed distance measure
is. It may be hard to conclude quantitative differences from this analysis as the
dispersion in the plot will not only reflect the number of reassortment events, but
also whether those events shuffle closely or distantly related lineages. In terms of
inter-lineage reassortment, the MDS in*
Figure 10
*does not add much to*
Figure 9
*(which shows each probable reassortment event) and meaningful quantification of
intra-lineage reassortment would require going beyond the MDS. We suggest cutting the
MDS. Also, results by Dudas, Bedford, Lycett, Rambaut, MBE, 2014 should be discussed;
the preprint has been available since March*.

We agree that the MDS may not accurately quantify the degree of reassortment. As
suggested, we have therefore removed MDS from our manuscript (including the figure), and
now directly use the phylogenetic trees of each gene segment to discuss our findings. In
addition, we have also included a discussion of the Dudas et al. paper. Their
observation that the PB2, PB1 and HA were genetically linked since the divergence of
Victoria and Yamagata lineages is consistent with our results. The new text in the
Results and discussion section is copied below:

“Phylogenies also suggest that the PB2 and PB1 gene trees (Figure 8) exhibit deep divergence, similar to the HA gene where
cocirculating viruses contain distinct Victoria and Yamagata genes. In contrast, the
other gene segments exhibit relatively recent divergence indicating that the prevailing
diversity of these genes originate from a single lineage. These results are consistent
with a detailed investigation of long term reassortment patterns of influenza B virus
lineages that revealed genetic linkage between the PB2, PB1 and HA protein genes.
Specifically, we observe that the PB2, PB1 and HA genes were consistently derived from a
single lineage, except for the short-lived subpopulation in 2004.”

*2) More direct measures of genetic diversity:*
Figure 3
*and*
Figure 8
*are labeled as “relative genetic diversity” (relative to what?).
We would like to see a more direct measure of diversity such as the average pairwise
distance in windows of a few months. We would also like to ask you to offer a clearer
interpretation of the GMRF results. The GMRF method gives smoothed estimate of the
rate of coalescence and is based on a neutral coalescent model. This measure is not
the same as genetic diversity and the meaning of estimates based on a neutral
coalescent is not clear*.

*The take home message from these plots seems to be that Vic has a
“burstier” and shallower phylogeny than Yam, with fewer lineages
carrying over from one season to another. This would show up as spikes and troughs in
the rate of coalescence and is consistent with the higher R0 and the smaller number
of introductions for Vic. If this is the desired interpretation, please discuss as
such*.

The term ‘relative genetic diversity’ (relative to itself through time) is
technically correct, appropriate, and has been used in many prior publications. We used
it in the place of ‘effective population size’ or
‘diversity’ because of the likely action of natural selection in shaping
the genetic structure of influenza virus, and to be consistent with prior publications
(e.g. Rambaut et al., 2008, Nature, 453: 615-9). We have elaborated on our usage of this
term in the Materials and methods section as follows:

“In the absence of natural selection (i.e. under a strictly neutral evolutionary
process) the genetic diversity measure obtained reflects the change in effective number
of infections over time (N_et_, where t is the average generation time).
However, because natural selection can play a major role in the evolution of influenza
HA, these are interpreted as ‘relative genetic diversity’, and which is
consistent with previous studies of influenza A virus (Rambaut et al., 2008, Nature,
453: 615-9).”

We agree with the astute interpretation of the GMRF plots, that the coalescent
represents the shape of the phylogeny. Accordingly, we have made extensive revisions to
the text in the Results and discussion section. Specifically, the increase and decrease
in ‘relative genetic diversity’ for Victoria signifies the genetic
bottleneck that occurs between seasonal epidemics, whereas for Yamagata diversity is
maintained between seasonal peaks and troughs, and which results in a gradual increase
of genealogical diversity through time. To make a better connection between the
coalescent analysis and the phylogeny, we have moved the description of the dated tree
(including Figure 4 of the revised manuscript),
to just below the GMRF description, within the section ‘Population dynamics of
influenza B virus’.

Furthermore, as suggested by the reviewers, we now provide an explicit measure of
changes in genetic diversity; the genealogical diversity (in years) (Figure 4 of the revised manuscript). The
genealogical diversity was measured by averaging the pairwise distance as time units on
the tree between random contemporaneous sample pairs. This measure is related to the
pairwise genetic diversity as measured on an accurate molecular clock-based phylogenetic
tree (Bedford et al., 2011, BMC Evol Biol 11: 220; Zinder et al., 2013, PLoS Pathog 9:
e1003104), and we believe is preferable to the pairwise methods proposed by the
reviewers, which do not account for phylogenetic structure and hence include extensive
pseudoreplication.

*3) Phylodynamic analysis: how were the reported years chosen? (please double
check the color code in Figure 2. 2011 seems to be fully dominated by Yam, but you
report Vic estimates in*
Figures 4 and 5*). The
discussion of the implications of higher R0 for Vic needs clarification. Prevalence
within one season depends on the start of the exponential growth phase, R0, and the
number of independent introductions (i.e., the number of lineages at the beginning if
the growth phase). The two strains differ in the latter two of these characteristics
in ways that affect prevalence in opposite ways. This needs more careful discussion:
the assertion that “these results are consistent with the more frequent
detection of Victoria lineage viruses during our sampling period (see Source data in
Dryad), indicating that Victoria viruses infected a higher proportion of the
population” (in the Results and discussion section) is a tautology (more
frequent detection corresponding to higher prevalence…). Would it be possible
to explicitly show that exponential growth of Vic and Yam differs between years? The
>5000 samples used for the age distribution should be sufficient to directly
show differences in dynamics if date information beyond year is available. If this is
not possible,*
Figure 6
*should be presented on a log scale and real incidence data should be
included*.

We apologize for the error in the color legend, where the names of Victoria and Yamagata
were swapped. We thank the reviewers for pointing this out. We have rectified this in
Figure 2 (of our revised manuscript) and
improved the colors to be consistent (red and black) with the rest of the
manuscript.

We have also revised the section on phylodynamics extensively, creating a new section
entitled ‘Transmission dynamics of influenza B virus’ (in the Results and
discussion section). As suggested, we have also included a cumulative case plot (in
semi-log scale) (Figure 5 of the revised
manuscript), using all positive cases for the same years that we estimated R_e_
(see Results and disscussion for the results of this analysis).

*4) Age distribution and glycosylation: Figure 16C shows odds ratios of for an
association between glycosylation at 212 and infections in age group 0-5. In Figure
16B, however, it would be much more useful to show that the fraction without 212
glycosylation correlates with the fraction of in age group 0-5. Otherwise, the
correlation is confounded by the total number of strain. When discussing the
structural implications of the observed sequences changes, more care should be taken
to distinguish hard evidence from previously solved structures (give PDB IDs and
references in figure or caption!), modeling, and hypothesis. Figure 15 could be
absorbed as a supplement to Figure 14. The “three major clusters” in
Figure 14C are not as clear as the text suggests. Elaborate or tone
down*.

This is a good question. We agree with the reviewers and therefore have switched to
fractions rather than absolute numbers (Figure 12 in the manuscript), although the confounding effect is the same for the
other age groups and the relative difference of the correlations could be compared. The
correlation with the youngest age group is still apparent, although weaker than before;
however, the difference to the other age groups remain as strong as before. Moreover, as
years with zero or one strain translate into extreme fractions (0 or 1) dominating the
correlation, we only show correlations for years with 10 or more Vic strains. The figure
legend has been adjusted accordingly.

We appreciate the comment on using previously resolved structures and their implications
to our findings and have taken great care when discussing them. In particular, we have
cleaned up our language to use strong wording only when we refer to the backbone
differences as observed in solved crystal structures for both lineages independently.
For visualization and comparison of general residue positions, such as for mutation
mapping (Figure 9 and Figure 11—figure supplement 1 of the revised manuscript),
we necessarily used homology models to account for the changes in the respective
sequences of each strain to be compared. However, these models are expected to be
structurally very close to the crystal structure templates simply due to the high
percentage identity (>90%) and lack of insertion/deletion changes. Importantly,
interpretation is made only at the level of rough structural residue positions (rather
than detailed side-chain conformations), which are expected to be reasonably accurate.
Furthermore, when adding ligands by superposition (Figure 11 in the manuscript), we computationally energy-minimized the
side-chains near the ligand to avoid any clashes following a successfully benchmarked
protocol (Krieger et al., 2009, Proteins 77 Suppl 9: 114-22). Finally, we have also
added the PBD IDs to all figures, and specified their exact use.

We have also toned down our description of the three major clusters in the legend of
Figure 11.

*5) Presentation and shortening: the manuscript is long with 16 figures in the
main text and its length dilutes the most important points. Also, please make every
effort to include parameters of programs used, scripts, and input files (such as
BEAST xmls)*.

We have made several changes to reduce the length of the manuscript (major changes
listed below), and also uploaded BEAST xml files to Dryad:

(a) The figure panel along with text that compares the phylodynamics between Australia
and New Zealand during 2002 was removed (Figure 6 in the original submission).

(b) We have combined multiple panels in Figure 5
(of the revised manuscript), including R_e_ estimates, comparison of
incidence/prevalence in 2008 (previously presented as a panel with (a)), and a new panel
showing cumulative cases as suggested in comment 3.

(c) We have considerably reduced the length of the section on reassortment by removing
any reference to MDS (and the figure) as suggested.

(d) We have removed the figure showing HA-NA site selection as suggested (Figure 11 in the original submission).

(e) We have moved a figure on structure to the figure supplement (Figure 11—figure supplement 1 in the revised manuscript),
as suggested.